# Fiber architecture in the human ventromedial striatum and its relation with the bed nucleus of the stria terminalis

Oliver Krüger[1,2*], Uwe Klose[1], Gisela E. Hagberg[2,3], Thomas Shiozawa-Bayer[4], Henry Evrard[5,6,7], Cintia Meszaros[1], Thomas Ethofer[3,8], Klaus Scheffler[2,3], Ulrike Ernemann[1], Benjamin Bender[1]

1 Department of Diagnostic and Interventional Neuroradiology, University of Tübingen, Tübingen, Germany, 2 High Field Magnetic Resonance, Max Planck Institute for Biological Cybernetics, Tübingen, Germany, 3 Department of Biomedical Magnetic Resonance, University of Tübingen, Tübingen, Germany, 4 Department of Anatomy, University of Tübingen, Tübingen, Germany, 5 Physiology of Cognitive Processes, Max Planck Institute for Biological Cybernetics, Tübingen, Germany, 6 Werner Reichardt Center for Integrative Neuroscience, University of Tübingen, Tübingen, Germany, 7 Center for Biomedical Imaging and Neuromodulation, Nathan S. Kline Institute for Psychiatric Research, Orangeburg, New York, United States of America, 8 Clinic for Psychiatry and Psychotherapy, University of Tübingen, Tübingen, Germany

* oliver.krueger@uni-tuebingen.de

## Abstract

The bed nucleus of the stria terminalis (BST) and the ventromedial striatum (consisting of the head of the caudate nucleus (hCN) and the nucleus accumbens (NAcc)) are both part of complex, foremost limbic networks involved in a variety of neuropsychiatric conditions. However, data on functional or structural connections between the BST and hCN in humans are scarce. In an earlier study using both diffusion tensor magnetic resonance imaging (DTI) and conventional histology we found a pathway from the BST to the orbitofrontal cortex apparently passing directly through the hCN. To confirm this finding, we now examined the hCN in human ex-vivo brain tissue using polarized light microscopy (PLM), a method particularly suitable for depicting myelinated nerve fibers. We further examined whether differences in fiber distribution inside the hCN could be depicted using high-resolution DTI data. PLM revealed different fiber populations inside the hCN and the NAcc. Fibers in the hCN were mostly related to the anterior limb of the internal capsule (ALIC) with some apparently terminating in the hCN while the majority exited the hCN to enter the prefrontal white matter. Fibers originating from the BST were only scarcely seen on this level and appeared to either terminate inside the hCN or join the ALIC. On levels below the anterior commissure, the BST strongly connected 1) to other basal forebrain structures including the NAcc, and 2) with the white matter of the medial prefrontal cortex. Differences in fiber density within the hCN could be reproduced on MRI data but with strong interindividual variation. In summary, PLM revealed a much more complex fiber architecture in the region of interest than suggested by our earlier DTI findings.

**Data availability statement:** Raw data of polarized light microscopy images and background images for correction as well as a compiled tool for the evaluation of polarized light microscopy images in the way used in this study are available for download under doi: 10.12751/g-node.xl2ctk. 14T MRI data of the tissue samples, anonymized 3T MRI data of the study participants, and masks underlying the analysis depicted in Figure 1 are available under doi: 10.12751/g-node.1lfvzi.

**Funding:** OK was financially supported by the medical faculty of the University of Tübingen (grant number F.1354230.1). We acknowledge support by Open Access Publishing Fund of University of Tübingen.

**Competing interests:** All authors declare no competing financial interests or potential conflicts of interest in relation to the work described. Outside of the submitted work BB has received consultancies from Medtronic (paid to institution) and is co-founder of AIRAmed GmbH.

**Abbreviations:** AC, anterior commissure; ALIC, anterior limb of the internal capsule; BA, Brodmann's area; BST, bed nucleus of the stria terminalis; DBS, deep brain stimulation; DTI, diffusion tensor magnetic resonance imaging; FA, fractional anisotropy; hCN, head of the caudate nucleus; mPFC, medial prefrontal cortex; NAcc, nucleus accumbens; OCD, obsessive-compulsive disorder; OFC, orbitofrontal cortex; PLM, polarized light microscopy; PTSD, post-traumatic stress disorder.

The study at hand shows that PLM can be a valuable tool for the verification of unclear or ambiguous DTI fiber tracking results.

---

## 1. Introduction

The bed nucleus of the stria terminalis (BST) and the ventromedial striatum (consisting of the head of the caudate nucleus (hCN) and the nucleus accumbens (NAcc)) are both part of complex brain networks comprising, among others, the medial prefrontal (mPFC) and orbitofrontal (OFC) cortices, the amygdala, and other limbic structures. They, respectively, are thought to be involved in a variety of psychiatric disorders, e.g., in post-traumatic stress disorder (PTSD), obsessive-compulsive disorder (OCD) and addictive behavior.

These conditions have been investigated by pre-clinical and clinical MRI methods. Several studies have reported changes in diffusion parameters [1] and metabolism [2], as well as reduced volume [3] of the caudate nucleus in PTSD. Meanwhile, a role of the BST for PTSD has been discussed on a cellular level in animal models [4,5], with mPFC-BST connections thought to play a critical role [6], and only in recent years, functional MRI findings have indicated altered BST activity in PTSD [7,8]. Regarding OCD, altered caudate nucleus and OFC/mPFC metabolism [9], as well as an altered functional interplay of the ventral caudate with mPFC and OFC [10] have been reported. In the last years, the BST has become a target of choice for deep brain stimulation (DBS) in cases of treatment-resistant OCD [11,12] with DBS being hypothesized to attenuate fear responses to ambiguous threats [13]. Regarding addiction, the caudate nucleus is considered to be of importance by mediating drug reinforcement and reward-seeking behavior [14,15], while the BST has repeatedly been hypothesized to play a key role in eliciting affective responses during drug withdrawal and hereby promote continuation of substance consumption [16,17]. Meanwhile, OFC and anterior cingulate cortex, regions important to higher-order executive and motivational functions and therefore to goal-directed activity, self-regulation, and impulse control, have been reported to show altered activity throughout different phases of addictive behavior [18,19].

However, to this date, data on functional and structural connectivity of the caudate nucleus and the BST in humans have been scarce. In an earlier study investigating the structural connections of the human BST by means of diffusion tensor magnetic resonance imaging (DTI) we observed a pathway from the BST to the OFC that appeared to directly pass through the head of the caudate nucleus [20]. In this study we were furthermore able to find evidence for the existence of such a pathway through histological examinations of a single human ex-vivo brain. Our observations were further supported by results from a DTI study of Kotz et al [21] that aimed at segmenting the caudate nucleus based on its diffusion properties.

In order to further substantiate our findings, we now examined the head of the caudate nucleus in four additional human ex-vivo brain hemispheres by means of polarized light microscopy (PLM). PLM is a technique widely used in mineralogy for the identification of mineral samples. Medical applications are found, for example, in

reproductive medicine (e.g., [22]), dentistry (e.g., [23]), dermatology (e.g., [24]), and further fields. In the context of neurosciences, there has been extensive work on developing techniques and mathematical methods for the acquisition, evaluation and depiction of myelinated fibers in the human brain (e.g., [25]). PLM is suitable for depicting myelinated nerve fibers with additional information on their three-dimensional course and without the need for any histological staining. Based on our novel findings we furthermore examined whether the spatial differences of fiber density inside the hCN can be outlined on high-resolution DTI data.

## 2. Materials and methods

### 2.1. Specimen for microscopy

The examined tissue samples were taken from two specimens of the Tübingen anatomy body donor program. The body donors were adjudged free from organic brain damage including dementia, cerebral ischemia, and cerebral bleeding. The brain was furthermore visually inspected for exclusion of advanced stages of atrophy. All body donors signed a written consent during lifetime permitting the use of their body and parts for science and teaching. The ethics committee of the University of Tübingen approved the anatomy body donor program (vote 237/2007BO1) as well as use of tissue from the body donor program for this polarization microscopy study (vote 771/2016BO2).

### 2.2. Preselection of the region of interest

For better orientation and exact localization of the regions of interest, the brains were first scanned at 3T (Siemens Magnetom PRISMA, Erlangen, Germany) using an MP2RAGE protocol with non-selective inversion pulses (TI1/TI2 = 700/2500ms; flip angle, FA = 4/5°; repetition time, TR = 7.7ms; echo time, TE = 3.16ms; volume TR = 5s; 0.8mm isotropic voxel size; GeneRalized Auto calibrating Partial Parallel Acquisition factor, GRAPPA = 3; partial Fourier factor = 6/8). The brain hemispheres were then separated along the sagittal midline. From each hemisphere, a block of approximately 5cm (rostrodorsal) x 2cm (craniocaudal) x 2cm (mediolateral) containing the anterior commissure (AC), the head of the caudate nucleus (hCN), and the anterior limb of the internal capsule (ALIC), as well as adjacent parts of the thalamus, the basal ganglia, the ventricles, and the medial prefrontal cortex was cut. The blocks were cut transversally with the cutting plane parallel to a line connecting the anterior and posterior commissures and reached from about 1cm above to 1cm below the AC. Before preparation for microscopy, DTI data of one tissue block (left hemisphere of specimen 2) were acquired with a 14T MRI Scanner (Bruker Biospin, Karlsruhe, Germany). For this purpose, a 3D segmented (N = 4) EPI sequence with TE/TR: 19.1/500ms; and 300µm isotropic voxels was used with b-values of 0 and 3500 s/mm$^2$. The diffusion gradients were applied along 64 non-collinear directions.

### 2.3. Preparation of tissue for polarization microscopy

For cryo-protection the tissue blocks were transferred into solutions of sucrose in 0.1mM PBS buffer with ascending concentration of sucrose up to 50%. The blocks were then embedded in a gelatin-albumin matrix. Finally, the tissue was cut into transversal slices with a thickness of 100 µm using a standard cryotome. To make later sorting of the slices easier, after each cut a photograph of the remaining block was taken with a DSLR camera installed above the cryotome. The slices were stored in PBS buffer and only mounted on glass slides and coverslipped immediately before microscopy.

### 2.4. Polarization microscopy of brain slices

The samples were examined at 25x magnification using a transmitted light microscope and a monochromatic CCD camera (Zeiss AxioCam HRm) with an image resolution of 1388x1040 pixels. In order to map the whole sample, we used a motorized table and the microscope's automated mosaicking function. To fit the needs of polarization microscopy the microscope was equipped with a narrow-band LED light source and an additional bandpass filter (light wavelength 635 nm ± 8 nm). Polarization filters were introduced into the light path with individually in-house manufactured holders.

For polarization microscopy, the examined samples are positioned between two linear polarizing filters introduced into the light path, the polarizer (sitting between the light source and the sample) and the analyzer (sitting between the sample and the microscope lens). The axis of the analyzer is arranged orthogonally to the axis of the polarizer so that, without any sample present between both filters, light transmission is reduced to a minimum. When linearly polarized light passes through a birefringent sample, like the myelin sheaths of nerve fibers, it will become elliptically polarized with the direction of the major axis depending on the angle between the axis of the polarizer and the main optical axis of the polarizing tissue, and only the component aligned with the analyzer will be detected by the camera. As an alternative, imaging can be performed using circularly polarized light by introducing a quarter-wavelength retarder that rotates together with the polarimeter into the light path.

For our study, images were acquired both with crossed polarizers only as well as with an additional quarter-wavelength retarder. Between acquisitions, while the sample remained stationary, all filters were rotated in steps of 10 degrees, starting from an initial setting of 0 up to 170 degrees (i.e., a semicircle). For each polarimeter setting, an additional set of empty background images was acquired for correction of background inhomogeneities.

## 2.5. Processing of microscopy images

Analysis of the microscope images was achieved using a MATLAB (Version 9.4, The MathWorks, Inc., Natick, Massachusetts, United States) script based on a simplified version (Matthias Valverde Salzmann, personal communication, September 2017) of the method described in great detail by Axer and colleagues [26,27]. The method was modified to increase contrast between white and grey matter and to accentuate in-plane directionality of the fibers at the cost of losing information on fiber inclination, since the fibers of interest in this study were expected to mostly run within the examined cutting planes.

## 2.6. MRI data acquisition in vivo

MRI data was collected from 12 healthy volunteers (4 females, 8 males; age (mean ± SD) 25.9 ± 3.8 years). Two participants were later excluded due to motion and other artifacts. The study was performed according to the Code of Ethics of the World Medical Association (Declaration of Helsinki) and was approved by the ethics committee of the University of Tuebingen. All participants gave their written informed consent prior to inclusion in the study. For each participant two DTI data sets were acquired, using a "Stejskal-Tanner" sequence with a b-value of 1000 sec/mm² (TR = 3000 ms, TE = 67 ms, 25 axial slices with slice 13 approx. at the height of the anterior commissure, voxel size 1.4 x 1.4 x 1.4 mm³, 64 directions of the diffusion weighted gradients spanning the whole sphere, 2 averages, intermingled volumes with a b-value of 0 sec/mm²), once with the phase encoding (PE) gradient oriented along the anterior-to-posterior direction and once with opposite polarity gradients (PE along the posterior-to-anterior direction).

## 2.7. MRI data processing and comparison with PLM

Diffusion-weighted data of both ex-vivo brain samples and in-vivo brain scans were preprocessed using FSL 5.0 (FMRIB Software Library, Oxford University, https://fsl.fmrib.ox.ac.uk/; [28]) including correction for participant movement and eddy current distortion using the images with opposite gradient polarity [29]. Further analysis of MRI data was then carried out using inhouse software.

For comparison with PLM results, masks of the hCN and the transition zone to the NAcc were created for each participant individually. For this purpose, we first selected overall 5 slices (3 above the AC, 1 at level of the AC and 1 below the AC) spanning a zone of 7 mm along the cranio-caudal axis. From each slice and for each hemisphere we then selected voxels belonging to the hCN, using fractional anisotropy (FA) maps for defining its borders with the ALIC, as well as b0 images to eliminate voxels belonging to the ventricles. To reduce partial volume effects of these structures we eliminated the outmost voxels of our masks. In the final analysis, each hCN mask was automatically segmented into 3 medial and 3

lateral segments, respectively (Fig 1a). The mean FA value was then calculated for each segment, both for the individual participants as well as for the whole group.

## 3. Results

### 3.1. Polarized light microscopy

Examination of the different brain hemispheres yielded overall similar results: Two major populations of fibers running preferentially in a rostrocaudal direction were identified in the head of the caudate nucleus (hCN) and the transition zone to the accumbens nucleus (NAcc).

1) Above and at level of the anterior commissure (AC) most fibers seen appeared as compact bundles with diameters of mostly 100 µm to 250 µm, with some bundles reaching diameters of up to 400 µm. Most of these bundles appeared to branch off the anterior limb of the internal capsule (ALIC; Fig 2a, yellow line) after its exit from the anterior thalamus, with several bundles running parallel to the ALIC (Fig 2a, red lines; Fig 3a–f) until reaching the rostral border of the hCN, where they often sharply turned laterally to rejoin the ALIC at its entry into the prefrontal white matter (Fig 3d–f). Other fiber bundles seemed to decrease in thickness throughout their course through the hCN

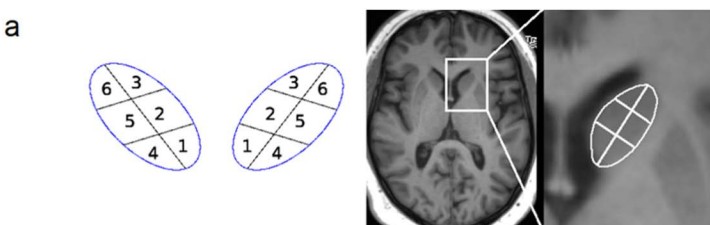

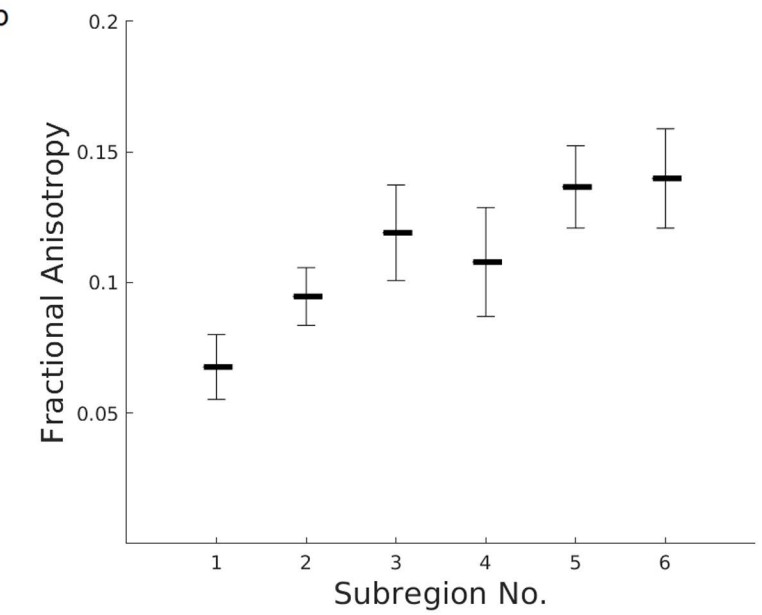

**Fig 1. Differences in fractional anisotropy in different hCN segments.** a) MRI masks of the hCN were automatically segmented into 3 medial (1-3) and 3 lateral (4-6) segments. b) For each of the segments, fractional anisotropy was averaged across all participants and slices. FA was seen to be lower in the medial segments 1 and 2 compared to the lateral segments 4 and 5 while no relevant difference was seen between the medial (3) and lateral (6) rostral segment.

until they were no longer definable, presumably terminating in the caudate (Fig 2a, green lines; Fig 3a–g). Apart from those fibers that definitely branched off the ALIC, some of the most medial ones were only distinguishable directly rostral of the BST, possibly originating there (Fig 2a, light blue lines; Fig 3a); however, no clear fiber structures were distinguishable in the supracommissural parts of the BST (Fig 2a, light blue line in schematic drawing). Generally, fiber density in the hCN was highest in the lateral two thirds, while in the medial third adjacent to the ventricle only few fibers could be identified.

2) Fibers below the AC differed from the ones above in that they were more loosely organized and less compact with typical bundle diameters of approximately 20 µm to 70 µm, and only a few with diameters greater than 100 µm (Fig 3g–h). They partly appeared to exit at the rostral border of the thalamus, but mostly originated in the infracommissural BST and neighboring basal forebrain structures. The lateral portion of those fibers entered the NAcc to terminate there, mostly in the posterior third (Fig 2b, light green lines; Fig 3h). The medial portion, however, passed medially of the hCN-NAcc-transition zone and the pericaudate (sub-) ependymal zone to enter the narrow medullary layer beneath the subgenual/subcallosal area (Brodmann's area (BA) 25) of the cingulate cortex (Fig 2b, dark blue lines; Fig 3h).

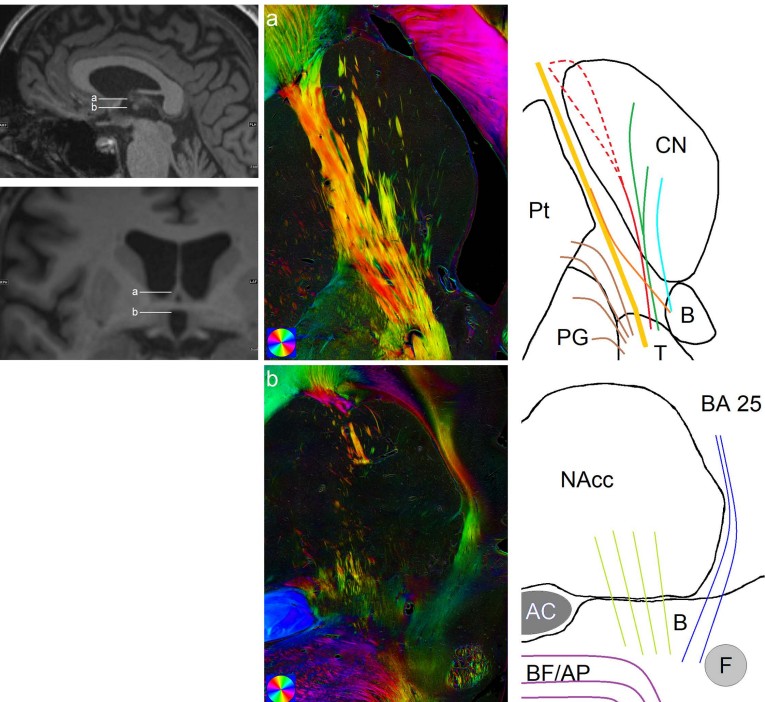

**Fig 2. Summary of main fiber populations in axial slices above and below the AC.** Polarization microscopy reveals multiple fibers with approx. rostrocaudal orientation in the vicinity of the head of the caudate nucleus (CN): **a)** The ALIC stretches from the anterior thalamus (T) to the prefrontal white matter (yellow bundle in schematic drawing). Branching off, there are fibers i) that reach the pallidal globe and putamen (PG and Pt; brown), ii) that terminate in the CN (dark green), iii) that run parallel to the ALIC in the lateral half of the CN to then sharply turn laterally at its rostral border and presumably rejoin the ALIC (red). Medial to those fibers leaving the thalamus are some fibers which start at the anterior border of the bed nucleus of the stria terminalis (B). Some of those appear to join the ALIC (orange) while others are directed into the CN (light blue). **b)** Below the AC, lateral to the fornix (F) and medial to the ansa peduncularis (AP; violet) in the basal forebrain (BF), multiple fine fibers can be seen in the gray matter of B and BF that cross under the AC, with their lateral portion (light green) terminating in the accumbens nucleus (NAcc) and their medial portion (dark blue) entering the medullary layer below the cortex of Brodmann's area 25 (BA 25).

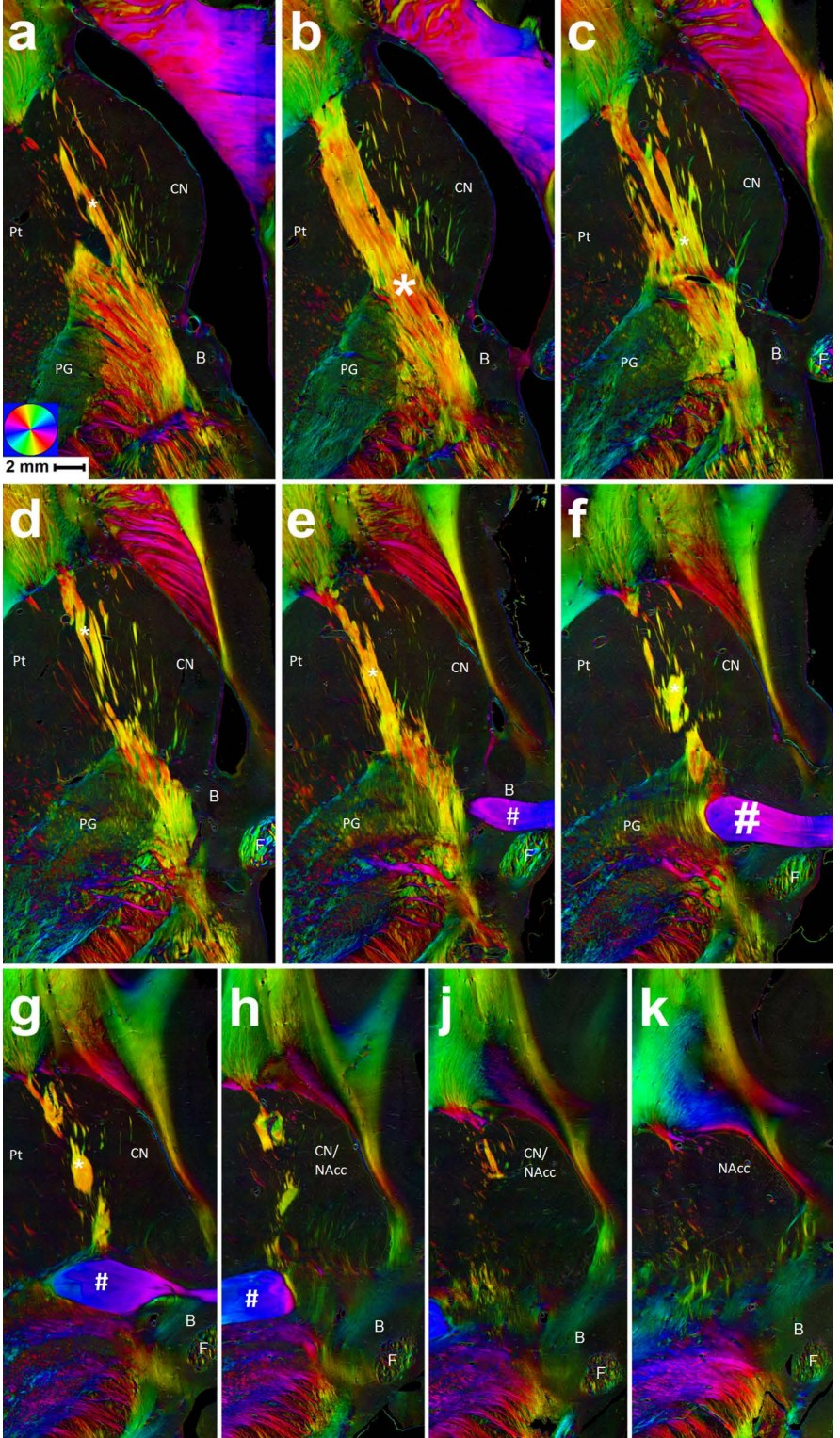

**Fig 3. Fiber populations throughout different axial slices above and below the anterior commissure.** 10 horizontal/transversal slices with a distance of 1 mm between each slice (i.e., every 10th slice), starting approx. 4 mm above the anterior commissure (#). Upper edge = rostral edge of slices;

left = lateral; right = medial; inferior edge = posterior. On some slices the ALIC (*) presents as a compact bundle (b, e) while on others it is loosened by stripes of grey matter (Pontes grisei caudatolenticulares) connecting caudate nucleus (CN) and putamen (Pt; a, c, d). Below the level of the interhemispheric part of the AC, the ALIC thins out, marking the transition zone of caudate and accumbens nucleus (CN/NAcc; h-j). B = bed nucleus of the stria terminalis; F = fornix; PG = pallidal globe.

### 3.2. Diffusion tensor MRI

For all examined slices and hCN segments, the across-participant average fractional anisotropy (FA) ranged between 0.07 and 0.14. Comparison of the left and right hemisphere of the across-participant average FA maps yielded two notable results. 1) Except for two of the overall 30 segments (i.e., the rostromedial segments of the two most ventral slices), the average FA in a hCN segment and its contralateral counterpart differed by not more than 0.01. 2) The average FA of the segments showed a mostly similar distribution for both hemispheres.

However, individual data was much less consistent, showing great interindividual differences for the comparison of single hCN segments and the FA distribution across the segments in each slice. Consequently, differences between the hCN segments mostly did not reach statistical significance in across-participant averages.

As mentioned above, across-participant averages of FA in the different hCN segments ranged between 0.07 and 0.14, meaning considerably less variation than would have been expected from PLM results, where some areas of the hCN showed an abundance of (approximately parallel) fibers, while other areas almost appeared to have no fiber content. Nonetheless, for the posterior and intermediate segments, FA was higher in the lateral half than in the medial half (Fig 1, segment 1 vs. 4 and 2 vs. 5). Furthermore, the medial intermediate segment (segment 2) showed higher FA than the medial posterior segment (segment 1).

In comparison to in-vivo DTI imaging, MRI of the ex-vivo brain samples at 14T allowed to better differentiate between the gray matter of the hCN and larger white matter fiber bundles within the hCN (Fig 4). Diffusion trace images allowed clear identification of thicker fiber bundles, but with hardly any inference on fiber directionality. FA differences at 14T between fibers in the hCN and the surrounding gray matter where less pronounced, while stronger fiber bundles of the adjacent ALIC could be clearly delineated. Still, FA differences between the medial and lateral parts of the hCN could even be detected with the bare eye. Meanwhile, FA maps of 3T in-vivo imaging only revealed unspecific visual inhomogeneities with no clear differences between medial and lateral parts of the hCN.

## 4. Discussion

In the current study we investigated nerve fibers in the transition zone of the head of the caudate nucleus (hCN) and the accumbens nucleus (NAcc) and their relationship to the bed nucleus of the stria terminalis (BST). For this purpose, we examined post-mortem tissue from 4 brain hemispheres using polarized light microscopy (PLM), a method particularly suitable for the examination of myelinated nerve fibers.

The microscopic examination of the hCN revealed different fiber populations. Above and at the level of the anterior commissure, fibers were organized as compact bundles mostly branching off the ALIC, with the majority re-joining the ALIC at the rostral end of the hCN. Furthermore, some fibers were delimited medial of the ALIC with possible origin in the supracommissural BST. Below the anterior commissure, fibers were more loosely organized with the lateral portion originating in the rostral thalamus and basal forebrain (including the infracommissural BST) and mostly terminating in the hCN-NAcc-transition zone, and the medial portion originating in the basal forebrain (including the infracommissural BST) and reaching into the medullary layer below the cortex of BA 25.

In order to determine whether differences between fiber distribution in different parts of the hCN could also be made visible in the living brain, we conducted high-resolution diffusion tensor magnet resonance imaging (DTI) in 12 healthy participants. DTI results varied greatly across participants but showed certain similarities with PLM results when looking at across-participant averages.

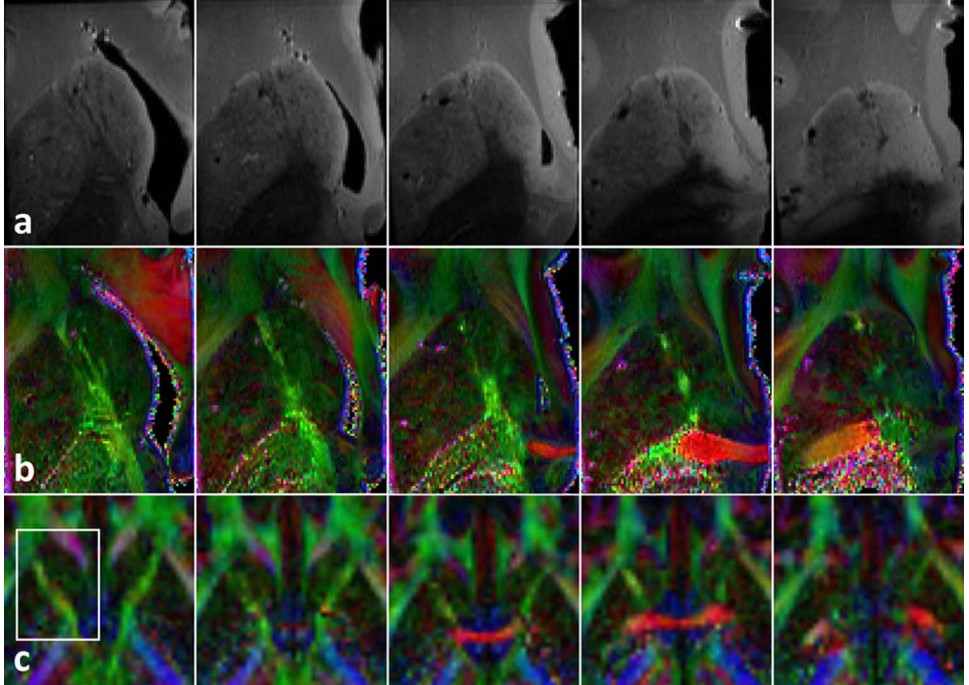

**Fig 4. Anatomical and DTI data of ex-vivo samples at 14 T compared to in-vivo DTI data at 3 T.** Axial sections at 14 T (a, b) and 3 T through the head of the caudate nucleus from superior to inferior with a slice gap of 1.4 mm reaching from approx. 3 mm above to 2 mm below the anterior commissure. 14T MRI allows the identification of larger fiber tracks within the hCN both based on diffusion trace **(a)**, as well as on fractional anisotropy **(b)**, and differences of FA between the medial and lateral parts are already visually detectable, especially in slices above the AC. 3T in-vivo imaging only reveals unspecific visual inhomogeneities on both FA maps **(c)** as well as trace (not shown) with no clear differences between medial and lateral parts of the hCN. The white box on the first 3T in-vivo image marks the sector displayed on 14T ex-vivo images.

### 4.1. Fibers above and at the level of the anterior commissure

The ALIC consists of several different fiber populations, most notably the anterior thalamic radiation, the frontopontine fibers, and fibers connecting striatum and frontal cortex. During embryonic development, the ALIC first develops as at least two separate parallel bundles between the laminae of the embryonic basal ganglia that only unite into a single bundle in later embryonic/early fetal development [30]. Even then, this single bundle remains interrupted by bridges of grey matter (Pontes grisei caudatolenticulares) connecting the caudate nucleus and putamen [31], giving this functional unit the well-known name "striatum". This circumstance probably partly accounts for those fibers seen to split from the bulk of the ALIC that run parallel to the ALIC through the lateral parts of the hCN and reunite with the ALIC at the rostral border of the hCN. PLM is not able to detect synapses and, therefore, it was impossible to determine whether those fibers were merely running through the hCN, or whether they were connected to hCN neurons which modulate signaling between thalamus and frontal cortex via the anterior thalamic radiation. This assumption is supported in general by the longstanding idea of cortico-thalamo-striatal loops [32,33] and more specifically by studies showing structural changes in [34,35], and altered connectivity between [36–38] those regions in patients with schizophrenia, as well as by an fMRI study showing increased BOLD signal in both thalamus and caudate nucleus after transcranial magnetic stimulation of the dorsolateral prefrontal cortex [39]. Given these implications for neural networks comprising both thalamus and CN, it was also not surprising to observe several of the fibers branching off the ALIC to not pass through the hCN, but to thin out until they likely terminated within the hCN.

A few of the medial-most fibers did not appear to branch off the ALIC, but instead to form at the rostral border of the BST. Those fibers were quite rare and hard to follow throughout the slices, since they did not run parallel to the slicing plane and

therefore were only traceable over a short range in each slice. While some of those fibers joined the ALIC, others most likely terminated in the posterior medial sector of the hCN. This appears to be in line with results from studies by Avery, Clauss [40] and Torrisi, O'Connell [41] which showed resting-state functional connectivity between BST and caudate nucleus, especially in these areas of the hCN. As explained above, BST and caudate nucleus both appear to be involved in a variety of neuro-psychiatric disorders like PTSD [2–5,42], OCD [10–12,43], and substance dependence [14–17]. However, their respective functional roles appear to differ regarding pathophysiology and symptoms of those disorders, and, to the authors' knowledge, a disturbance of this functional interplay between BST and hCN in the context of specific disorders to this date has only been described in a single study comparing PTSD patients with healthy controls [8], and it was only seen between the BST of one, and the hCN of the contralateral hemisphere. It thus remains speculative what the exact functional implications of these connections might be. It furthermore has to be taken into consideration that the DTI results reported by Kotz, Anwander [21] and ourselves [20] may rather be explained by the high fiber content within the hCN as a result of ALIC-related bundles, with the close proximity of the BST to the hCN, the small size of the BST of a few millimeters, and partial volume effects due to the low resolution of the DTI data accounting for the seen fiber trajectories.

### 4.2. Fibers below the anterior commissure

The finding of fibers originating from the BST and adjacent basal forebrain structures that connect to the NAcc, which were microscopically seen in abundance, is well in line with anatomical knowledge obtained from different animals (e.g., [44–48]) as well as from functional implications of the extended amygdala (e.g., [40,41,49–51]). Beyond these results, we were furthermore able to distinguish fibers exiting the BST/basal forebrain that ran medially of the hCN-NAcc-transition zone to enter the white matter of BA 25, the most posterior area of the ventromedial prefrontal cortex. As delineated below *(Methodical considerations)*, the methods used in this study are limited concerning resolution and the detection of synapses, and therefore, we were not able to determine where those projections entering the medullary layer of BA 25 terminate. It is standing to reason that at least a part locally innervates neurons of BA 25, corresponding to various reports of connectivity between the BST and the rodent equivalent of BA 25, the so-called infralimbic cortex [45,52]. Furthermore, connections between BST and the prelimbic cortex, which is rostrally adjacent to the infralimbic cortex in rodents and supposed to be equivalent to BA 32 [53], have been described as well [45,54], although less frequently. However, in human and non-human primate studies there are indications of the BST being connected to pre- and orbitofrontal cortex regions even further rostral [55,56], and a role of these connections has been assumed in the pathological processes of social anxiety disorder (see our discussion in [20]).

### 4.3. Comparison of DTI and PLM results

Our in-vivo DTI experiment was carried out to test whether differences of fiber densities in different parts of the hCN could be resolved at high resolution. To this end we looked at fractional anisotropy (FA), a parameter that allows inference on the directionality of diffusion within a voxel. After averaging both across participants and selected slices we made two main observations: 1) Apart from the most rostral third of the hCN (segments 3 and 6, Fig 1), FA was higher in its lateral half than in its medial half, and 2) for the medial segments FA appeared highest in the rostral third (segment 3), and lowest in the posterior third (segment 1) of the hCN (Fig 1). Observation 1 is well in line with our PLM findings, showing low fiber content in the medial aspects of the hCN, especially in its intermediate third, while in its rostral third fibers were seen to run numerously both through the lateral and medial half (Fig 3, b–e). In order to interpret observation 2, especially the difference between the posterior and intermediate medial segments, it has to be taken into consideration that FA is not a direct measure of fiber density but primarily consists of information about directionality of diffusion, meaning that fibers with different orientation within a voxel will lead to a decrease in FA. While in the intermediate and rostral segments the observed fibers were oriented almost parallelly, they were stretched out at wider angles in segment 1, resulting in a decrease in FA despite a similar or even higher fiber content than in segment 2.

The extent of these effects varied across the different examined slices. This may, on one hand, be an effect of actual differences in fiber distribution. On the other hand, the position of the five ROI slices varied slightly between the different participant data sets.

Although group analysis revealed the abovementioned similarities to PLM data, we observed great inter-individual variation of diffusional properties in the different slices and segments. PLM, on the other hand, yielded mostly similar results in all examined hemispheres. Several methodological aspects might account for these differences, e.g., the effect of (slight) movement of participants during the scan, differences in the exact orientation of image planes, and resulting inconsistencies in the definition of the ROIs. Furthermore, there were pronounced differences in the size of the MRI participants' caudate nuclei, resulting in a lower number of voxels per segment in some of the participants, which might account for outliers. Moreover, it has to be considered that there might actually be great inter-individual variation in fiber distribution across the hCN that was not observed in our microscopy experiment because of the relatively low number of brain hemispheres examined with PLM. However, considering the abovementioned embryonic development of the ALIC, we would generally expect the fiber content in the more lateral parts of the hCN to be higher than in its medial segments. Finally, the average age of the MRI study participants was much lower than that of the body donors; however, we did not expect this fact to lead to relevant differences in fiber architecture considering that body donors were adjudged free from advanced neurodegenerative diseases, the selected brains showed no relevant atrophy, and post mortem MRI scans of the selected brains did not indicate relevant white matter damage. DTI of the tissue samples at 14T revealed strong fractional anisotropy in bigger fiber bundles and even allowed identification of larger fibers within the gray matter of the hCN. However, since DTI resolution was still relatively low compared to PLM, further differentiation of those fibers was very limited (Fig 4).

## 4.4. Methodical considerations

There are several methodical limitations of the applied methods of polarized light microscopy (PLM) and diffusion tensor magnetic resonance imaging (DTI) to be considered when interpreting the results of this study.

A major issue when examining fiber pathways by means of DTI is its low resolution compared to microscopy, but also other anatomical MRI sequences. In our recent experiment we were able to increase resolution to 1.4 mm isotropic with an acceptable signal-to-noise ratio. This, compared to our earlier study at a resolution of 2.0 mm isotropic, leads to a considerably better delineation of intermediate fiber bundles of a few millimeters' diameter, like the anterior commissure or the anterior limb of the external capsule. Thinner fiber structures, like the external and extreme capsules, on the other hand, could still not be sufficiently differentiated from bordering structures, like the claustrum. Still, strong fiber bundles or several bundles of similar orientation may strongly influence diffusional properties of the voxel they are contained in.

Microscopy, compared to MRI, allows for much higher resolution. In our case, even when only using 25-fold magnification, one camera pixel theoretically corresponded to an image detail of 2.5 μm; however, the actual optical resolution of the microscope at this magnification is limited to a little below 10 μm under ideal conditions. Considering usual diameters of 0.2 μm – 10 μm for single myelinated nerve fibers in the corpus callosum [57], it seems natural to use higher resolution objectives. The actual limitation to resolution in regard to information on fiber orientation, however, is posed by the thickness of the examined slices which, at 100 μm, was one order of magnitude higher than the in-plane resolution. Therefore, in order to gain more detailed information on fiber orientations, slice thickness needs to be reduced. This, however, entails further methodical challenges. Thinner slices are more difficult to handle in a way that they get both damaged and distorted more easily. They furthermore result in an obvious substantial increase in overall measurement time per brain sample, considering that with the experimental setup used, the average acquisition time for each 2 cm x 4 cm slice ranged around 2–3 hours. Finally, a reduction of slice thickness leads to reduced myelin content per pixel, and therefore to a less pronounced effect on the polarization state of the transmitted light, resulting in lower amplitude variations of the signal over different polarimeter angles and, therefore, in less precise estimations of fiber orientation [25]. For the questions examined in this study, we found the used resolution to be sufficient, since the main goal was to search for correlates

of our earlier described, MRI-based results [20]. However, the issue of resolution may gain importance when examining internal connectivity of smaller, more complex structures, like different nuclei of the thalamus or the amygdala.

Apart from overall fiber content and orientation, the extent of myelination of nerve fibers in the brain possibly plays an important role in the integrity of signaling pathways and their disruption and malfunction, especially in the light of recent insights on the role of myelin sheaths in the plasmon-polariton model of signal transduction [58]. As mentioned, due to limited resolution, our experimental setup primarily allows identification of fiber bundles, but not of individual fibers within these bundles. Furthermore, since no staining is applied on the tissue samples, the method is not feasible for the delineation of membranes and subcellular structures, which also limits information on myelination density and especially thickness of myelin sheaths. Since the polarization effect of a myelinated fiber bundle depends both on the amount of myelination as well as the angle of the fiber in regard to the cutting plane of the sample, the resulting signal always carries information about both parameters at the same time. There have been different approaches to estimating fiber inclination either using a setup with a tilting stage [59] or mathematical estimation of fiber inclination [60]. While the first method might allow inference on myelination density in a second step, it is not easily applicable to our experimental setup. The second method, on the other hand, depends on hypothetical estimates of fiber density in the first place and therefore appears mainly suitable for examination of overall larger scale samples and less for smaller, complex structures (the basal ganglia are mentioned in that work specifically as a challenging region for the presented technique). Nonetheless further advances in this technique might later allow a re-evaluation of the data acquired for this study in regards to myelination density.

One limitation inherent to both PLM and DTI is the inability to display synapses. Therefore, especially in areas with high fiber content, it often is not possible to determine the end of a specific fiber bundle as mentioned above in regard to the nerve fibers entering the medullary layer of BA 25. For PLM, this issue could at least partly be overcome by additional examination of slices after staining for synapses. However, this would again require a significant increase in resolution.

## 5. Conclusion

In the current study, we examined myelinated fiber bundles in the head of the caudate nucleus and the accumbens nucleus in the brains of two human body donors by means of polarized light microscopy (PLM). Corresponding to our results we then segmented the hCN based on properties derived from DTI in vivo.

Using PLM, we identified fiber bundles with diameters of mostly 100 µm to 250 µm originating from the anterior limb of the internal capsule with a portion completely traversing the hCN to rejoin the ALIC and the other portion terminating in the hCN. Above the anterior commissure, medial to those fibers, we saw several fibers forming at the rostral border of the BST. Some of those joined the ALIC, while others appeared to terminate in the posteromedial caudate head, consistent with studies on resting-state functional connectivity between BST and hCN. Below the AC, we identified finer fiber bundles with diameters below 100 µm originating from the basal forebrain, including the infracommissural BST; the lateral portion of those fibers reached into the NAcc and appeared to terminate there; the medial portion on the other hand passed medially to the hCN/NAcc transition zone into the medullary layer of the subcallosal cingulate cortex (BA 25). While we were not able to conclusively verify our earlier, MRI-based findings of a strong fiber bundle from the BST through the hCN into the prefrontal white matter [20] we still succeeded in finding morphologic correlates both for the earlier described diffusional anisotropy in the hCN and for DTI fiber tracking results showing structural connectivity between BST and prefrontal cortex.

Our recent DTI measurements, which were performed at a higher resolution than in previous studies [20,21], revealed a certain degree of diffusional anisotropy in the hCN as well as a gradient between the lateral and medial aspects of the hCN. More precisely, FA appeared to be lower in the medial parts of the hCN, corresponding with our PLM findings that many fibers in the hCN were related to the ALIC and therefore concentrated laterally in the hCN. However, diffusional properties inside the hCN showed great interindividual variance.

The study at hand shows that DTI and derived fiber tracking results, which are often used to show structural connectivity between different brain regions of interest in neuroimaging studies, still have to be interpreted cautiously, especially when it comes to new, unexpected findings. PLM, despite its limitations, appears to be a suitable tool to further evaluate ambiguous or unclear DTI results.

## Supporting information

**S1 Data. FA Data.**
(ZIP)

## Author contributions

**Conceptualization:** Oliver Krüger, Uwe Klose, Thomas Ethofer.

**Data curation:** Uwe Klose.

**Formal analysis:** Oliver Krüger, Uwe Klose, Gisela E. Hagberg.

**Funding acquisition:** Oliver Krüger.

**Investigation:** Oliver Krüger, Gisela E. Hagberg.

**Methodology:** Uwe Klose, Gisela E. Hagberg.

**Resources:** Thomas Shiozawa-Bayer, Henry Evrard, Klaus Scheffler, Ulrike Ernemann.

**Software:** Oliver Krüger, Uwe Klose, Gisela E. Hagberg.

**Supervision:** Ulrike Ernemann, Benjamin Bender.

**Validation:** Cintia Meszaros, Benjamin Bender.

**Visualization:** Oliver Krüger.

**Writing – original draft:** Oliver Krüger.

**Writing – review & editing:** Uwe Klose, Gisela E. Hagberg, Cintia Meszaros, Benjamin Bender.

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
