## [Decision Letter · Decision Letter 0]

12 Nov 2024

PONE-D-24-36525Fiber architecture in the human ventromedial striatum and its relation with the bed nucleus of the stria terminalisPLOS ONE

Dear Dr. Krueger,

Thank you for submitting your manuscript to PLOS ONE. After careful consideration, we feel that it has merit but does not fully meet PLOS ONE’s publication criteria as it currently stands. Therefore, we invite you to submit a revised version of the manuscript that addresses the points raised during the review process.

We look forward to receiving your revised manuscript.

Kind regards,

Stijn Michielse, PhD

Academic Editor

PLOS ONE

Journal Requirements:

2. Thank you for stating the following financial disclosure: “OK was financially supported by the medical faculty of the University of Tübingen (grant number F.1354230.1). We acknowledge support by Open Access Publishing Fund of University of Tübingen.”

3. Thank you for stating the following in the Competing Interests section: “All authors declare no competing financial interests or potential conflicts of interest in relation to the work described.

Outside of the submitted work BB has received consultancies from Medtronic (paid to institution) and is co-founder of AIRAmed GmbH.”

Please confirm that this does not alter your adherence to all PLOS ONE policies on sharing data and materials, by including the following statement: "This does not alter our adherence to PLOS ONE policies on sharing data and materials.” (as detailed online in our guide for authors http://journals.plos.org/plosone/s/competing-interests). If there are restrictions on sharing of data and/or materials, please state these. Please note that we cannot proceed with consideration of your article until this information has been declared. Please include your updated Competing Interests statement in your cover letter; we will change the online submission form on your behalf.

4. We note that you have indicated that there are restrictions to data sharing for this study. PLOS only allows data to be available upon request if there are legal or ethical restrictions on sharing data publicly. For more information on unacceptable data access restrictions, please see http://journals.plos.org/plosone/s/data-availability#loc-unacceptable-data-access-restrictions. Before we proceed with your manuscript, please address the following prompts: a) If there are ethical or legal restrictions on sharing a de-identified data set, please explain them in detail (e.g., data contain potentially identifying or sensitive patient information, data are owned by a third-party organization, etc.) and who has imposed them (e.g., a Research Ethics Committee or Institutional Review Board, etc.). Please also provide contact information for a data access committee, ethics committee, or other institutional body to which data requests may be sent. b) If there are no restrictions, please upload the minimal anonymized data set necessary to replicate your study findings to a stable, public repository and provide us with the relevant URLs, DOIs, or accession numbers. For a list of recommended repositories, please see https://journals.plos.org/plosone/s/recommended-repositories. You also have the option of uploading the data as Supporting Information files, but we would recommend depositing data directly to a data repository if possible. We will update your Data Availability statement on your behalf to reflect the information you provide.

Additional Editor Comments:

Thanks for the work in putting this manuscript together. It is overall well written and requires minor revisions.

See the reviewer comments and address them accordingly.

Reviewers' comments:

Reviewer's Responses to Questions

**Comments to the Author**

1. Is the manuscript technically sound, and do the data support the conclusions?

Reviewer #1: Yes

Reviewer #2: Yes

2. Has the statistical analysis been performed appropriately and rigorously? 

Reviewer #1: Yes

Reviewer #2: N/A

3. Have the authors made all data underlying the findings in their manuscript fully available?

Reviewer #1: Yes

Reviewer #2: Yes

4. Is the manuscript presented in an intelligible fashion and written in standard English?

Reviewer #1: Yes

Reviewer #2: Yes

5. Review Comments to the Author

Reviewer #1: This paper utilized polarized light microscopy (PLM) to study their earlier discovery that a pathway from the BST to the orbitofrontal cortex apparently passing directly through the hCN. Moreover, the high-resolution DTI data is used to analyze the differences in fiber density inside the hCN. The applied limitations of PLM and DTI are discussed to show that PLM can be a valuable tool for the verification of unclear or ambiguous DTI fiber tracking results. There are some improved points as follows as

(1) In the Introduction, it is lacked that the current status and application of PLM in brain research. The PLM is primarily used in crystallography for geologists, mineralogists, and chemists. However, in the past few years, the PLM has been employed in biology.

(2) In the 2.4 section, Is your PLM an off-the-shelf instrument or a modified instrument? In the second paragraph, the words “The axis of the upper filter (called analyzer) is arranged orthogonally to the axis of the lower filter (called polarizer) so that, without any sample present between both filters, light transmission is reduced to a minimum” could be confusing. The sample should be placed between both filters. The light first passes the Polarizer, second passes the Birefringent Specimen, and finally passes the Analyzer.

(3) In Results, the caption of Fig.4 is incomplete.

(4) Based on the PLOS Data policy, authors should submit the following data: 1. The values behind the means, standard deviations and other measures reported; 2. The values used to build graphs; 3. The points extracted from images for analysis. So the values of Fig.1(b) may be submitted.

Reviewer #2: The submission concerns a microscopic study of samples of neural tissue from a selected region of CNS. The architecture and organization of white matter in these samples has been observed by the microscope and supported by the MRI of related region. The multi fiber bundles in white matter of diameter 100-250 micrometer were observed. However, neither diameter distribution of particular fibers in bundles, nor the thickness of myelin sheath for particular fibers were assessed. These parameters are of primary significance for the speed of signaling through these paths. Such observations would be beneficial especially in view of recent advances in understanding of the fast saltatory conduction in myelinated axons, which conditions the architecture and organization of neural paths in white matter (including those considered in the submission). The conventional cable model is unable to properly describe signaling in white matter of CNS as the ion diffusion speed is too slow in human axons and the other mechanism of the stimulus in myelinated axons must be considered. The role of myelin is also different than previously thought upon the cable model (cf. e.g., :https://doi.org/10.1103/PhysRevE.109.034401, Neuroscience 505 (2022) 125–156).

Despite the limited range of the study, the paper is probably worth publication. The manuscript is well organized and carefully written. It might be published provided some revision and the supplementation with comments addressed to functionality of studied structures is added, taking into account a present status in understanding of action potential transduction in myelinated axons. The architecture and local organization of white matter is the function of signal transduction mechanism, its velocity, inter-axonal interaction and so on. Some comment towards such problems would enhance the significance of microscopic observations presented in the paper.

6. PLOS authors have the option to publish the peer review history of their article (what does this mean? ). If published, this will include your full peer review and any attached files.

**Do you want your identity to be public for this peer review?** For information about this choice, including consent withdrawal, please see our Privacy Policy .

Reviewer #1: No

Reviewer #2: No

---

## [Author Response · Author response to Decision Letter 0]

24 Mar 2025

Responses to points raised by the editor:

1. We reformatted the manuscript to meet formatting requirements.

3. All authors declare no competing financial interests or potential conflicts of interest in relation to the work described. Outside of the submitted work BB has received consultancies from Medtronic (paid to institution) and is co-founder of AIRAmed GmbH. This does not alter our adherence to PLOS ONE policies on sharing data and materials.

4. We have made all data used for this study available for download via data repositories. Please update our Data Availability statement as follows: “Raw data of polarized light microscopy images and background images for correction as well as a compiled tool for the evaluation of polarized light microscopy images in the way used in this study are available for download under doi: 10.12751/g-node.xl2ctk.

14T MRI data of the tissue samples, anonymized 3T MRI data of the study participants, and masks underlying the analysis depicted in Figure 1 are available under doi: 10.12751/g-node.1lfvzi.”

In addition, the single FA values extracted from the abovementioned masks are now provided as supporting data in text format (FA_data.zip) with one file for each extracted segment of each participant (Sub_no_Seg_no_FA), one file for each participant containing mean FA values for the six different segments (Sub_no_mean_FA), as well as one file with above participant averages for the six different segments (Mean_FA_values).

5. To the best of our knowledge, our reference list does not contain articles that have been retracted.

Responses to points raised by reviewer 1:

1. We added a paragraph on the use of polarized light microscopy in medical applications to our introduction.

2. Polarization filters were introduced into the light path with an individually in-house manufactured holder. The sentence on the placement of the different filters was reworded for better understandability.

3. We have added additional information to the caption of Fig 4.

4. The individually extracted ROIs from our participants’ FA maps are now part of our data repository under doi: 10.12751/g-node.1lfvzi. The extracted FA values for each participant and segment have furthermore been uploaded as supporting data.

Responses to points raised by reviewer 2:

We acknowledge that the extent of myelination of nerve fibers in the brain plays an important role in the integrity of signaling pathways, especially in the light of recent insights on the role of myelin sheaths in the plasmon-polariton model of signal transduction. Unfortunately, the technique we used in the study at hand is limited in this regard. We have dedicated an additional segment to address these limitations in our paragraph on Methodical considerations.

---

## [Editor Report · Decision Letter 1]

2 Apr 2025

Fiber architecture in the human ventromedial striatum and its relation with the bed nucleus of the stria terminalis

PONE-D-24-36525R1

Dear Dr. Krueger,

We’re pleased to inform you that your manuscript has been judged scientifically suitable for publication and will be formally accepted for publication once it meets all outstanding technical requirements.

Kind regards,

Stijn Michielse, PhD

Academic Editor

PLOS ONE
---

## [Editor Report · Acceptance letter]

PONE-D-24-36525R1

PLOS ONE

Dear Dr. Krüger,

I'm pleased to inform you that your manuscript has been deemed suitable for publication in PLOS ONE. Congratulations! Your manuscript is now being handed over to our production team.

Kind regards,

on behalf of

Dr. Stijn Michielse

Academic Editor

PLOS ONE